# Passification-Based Robust Phase-Shift Control for Two-Rotor Vibration Machine

**Boris Andrievsky** [1,2,*], **Iuliia Zaitceva** [1,2,3] **and Itzhak Barkana** [4]

1   Control of Complex Systems Laboratory, Institute for Problems in Mechanical Engineering of Russian Academy of Sciences (IPME RAS), 61 Bol'shoy Pr. V.O., 199178 Saint Petersburg, Russia
2   Department of Applied Cybernetics, Faculty of Mathematics and Mechanics, Saint-Petersburg State University, Stary Peterhof, Universitetsky Prospekt, 198504 Saint Petersburg, Russia
3   Department of Automatic Control Systems, Saint Petersburg Electrotechnical University "LETI", Professora Popova Str., 5, 197376 Saint Petersburg, Russia
4   BARKANA Consulting, Ramat-Hasharon 4720937, Israel
*   Correspondence: b.andrievsky@spbu.ru; Tel.: +7-812-321-4776

**Abstract:**   In this paper, the solution to the problem of robust control of the phase shift during rotation at a given speed of the unbalanced rotors for a two-rotor vibratory machine is presented. The solution to this problem is relevant for the development of vibration technologies (for example, a vibro-transportation of bulk materials). The proposed controller includes two proportional-integral (PI) rotor speed controllers with a cross-coupling, which receive signals with opposite signs from the phase shift controller. Unlike previous works, where a PI controller for phase shift control was also taken, including the adaptive controller with an implicit reference model (IRM), in the present paper, a relay-type signal controller with an integral component without a parametric adaptation is used. This approach allows, while maintaining robustness, to increase the operation speed and accuracy of the control process, avoiding at the same time the possible divergence of the tunable parameters due to the influence of noises and disturbances caused, among other things, by vibrations of the setup's structural elements and measurement errors. For the control law design, the speed-gradient method was employed. For various types of reference phase-shift signals (constant, harmonic, chaotic), the results of extensive experimental studies performed on the mechatronic vibration setup and the simulations accomplished based on the results of identifying the parameters of the stand drive model are presented in the paper. The obtained results confirm the efficiency and robustness of the proposed algorithm and allow one to reveal the system performance properties.

**Keywords:**  vibration machine; unbalanced rotors; mechatronics; phase shift; speed-gradient; relay control; robustness; nonlinear control

## 1. Introduction

Vibration technologies are used in many areas of industry and production: in the processing, metallurgical, machine-building, and chemical industries, in the production of building materials, and in machines for grinding, fine grinding, or surface treatment of various parts. Vibratory machines have found applications for a huge range of tasks, for example, in the construction and production of building materials for compacting concrete mix, soil and road surfaces for the formation of reinforced concrete products, immersing piles in the ground; in mechanical engineering, the mining industry for drilling, loading, and delivery of rock mass, screening; in transport for unloading packed materials, bulk materials; in agriculture as separators, vibratory pumps, and tillage devices. The property of self-synchronization of the speeds of rotation of several kinematically and electrically unbalanced rotors is used in machines such as conveyors, feeders, screens, crushers, and mills; see [1–7] and the references therein.

The control of the phase shift between the two rotors of a vibrating machine in order to obtain various types of trajectories of movement of the working platform is essential in vibration technology and technologies built on its basis. Directed, on average, the movement of a particle along an oscillating rough surface is the asymmetry of the form of oscillations due to the inequality of the time intervals between the acceleration extrema, known as the temporal asymmetry of the oscillations of the working surface; see [5,6,8]. The results obtained are important for the design of vibro-transporters. Changing the vibro-transportation parameters is traditionally carried out using the settings of the vibration drives, for example, by the hand-made adjustment of the eccentric masses positions. An advanced step in the implementation of intelligent technological systems is the use of the principles of mechatronics by the creation of computer-controlled vibratory systems with feedback; see [9–12]. In [9], the experimental results obtained at the Multi Resonance Mechatronic Laboratory Setup (MMLS) SV-2M are presented, showing the efficiency of the proposed approach. In a recent paper, Andrievsky and Zaitceva [12] demonstrated an application of the phase-shift control for the chaotization of the platform vibrations. The approach of [9,10] has been extended by implementing simple adaptive control (SAC) with an implicit reference model (IRM) in [11], where the possibility of efficiently applying the SAC method to a real technical system is shown. The algorithm obtained in [11] refers to the velocity gradient (SG) algorithms, cf. [13–17] and Section 2.2 below, with the implementation of the controller parameters tuning. However, in real problems, in the presence of unmodeled dynamics, disturbances (for example, caused by the platform vibrations), noises, measurement errors, and data sampling, the parameters of the adjustable controller can leave the allowable zone. Reducing the gain of the adaptation algorithm in combination with parametric feedback in it, as well as anti-windup correction, reduces the achievable speed of the system and its accuracy in the tracking mode. Therefore, in this work, another type of SG algorithm is developed and studied, namely, a robust signal algorithm, where adjustment of the controller parameters is not employed. This approach allows, while maintaining robustness, to increase the speed and accuracy of the control process while avoiding a possible divergence of the controller parameters.

In this paper, we also use new results that simplify the stability analysis and eliminate some demanding continuity conditions that are usually considered to be necessary for the stability of the system under analysis. Although most publications base their stability analysis on Barbalat's Lemma, which seems to make continuity the necessary condition, recent publications present a new theorem of stability, which is a direct extension of the original Lyapunov theorem of stability for the case when the Lyapunov derivative is only negative semidefinite, and they managed to show that continuity is not needed for stability.

The remainder of the paper is organized as follows. The mathematical preliminaries are given in Section 2, where the stability analysis is discussed and the Speed-Gradient (SG) design method in the form used for the phase-shift control law design is presented. Section 3 presents a brief description of the laboratory setup SV-2M used for carrying out the experimental research. The control law for synchronization and phase-shift control of the two-rotor vibration machine is presented in Section 4. Section 5 gives a detailed exposition of the numerical examination and experimental results for SV-2M control. Concluding remarks and the future work intentions in Section 6 finalize the paper.

## 2. Mathematical Preliminaries

### 2.1. Stability Analysis

To describe the dynamics of the controlled plant, the following equations in the state-space form are used

$$\dot{x}(t) = f(x, \theta, t), \tag{1}$$

where $x(t) \in \mathbb{R}^n$ is the plant state vector; $\theta(t) \in \mathbb{R}^m$ denotes control vector (the input vector).

Important Note: Like almost all publications and as a result of using Barbalat Lemma, until now, we used to write: "$f(\cdot)$ is a vector-function continuous in $x$, $\theta$, $t$, continuously differentiable in $\theta$."

Barbalat Lemma was the result of some counterexamples that seem to show that a function may reach a finite limit as time tends to infinite, and yet, its derivative may keep going up-and-down forever. Therefore, at its time, Barbalat's Lemma was important because it allowed some analysis of system stability.

It is useful to reproduce the Lemma, which in one of its formulations says: if the function $f(t)$ has a finite limit as time tends to infinite and if its derivative $\dot{f}(t)$ is uniformly continuous, then $\dot{f}(t)$ tends to a zero limit as time tends to infinity.

As we already mentioned, this was important at the time, as it allowed one to have some proof of stability. However, these continuity conditions raise doubts about the guarantee of stability of real-world systems, which contain nonlinearities, because strict continuity cannot be guaranteed in real-world applications, and therefore, this condition seems to imply that any occasional discontinuity may negatively affect the stability of the system.

Fortunately, some late results of LaSalle [18] seemed to mitigate these conditions. However, any transient term that may affect the strict negativity of the Lyapunov derivative seems to also affect the stability analysis. Furthermore, using the contribution of Arzstein [19] to this topic and further developing those first contributions, new results (see [20–26]) allow to eliminate these tough continuity conditions and thus add to the guarantee of stability of applications. Furthermore, a careful review of such delicate concepts as limit and derivative limit shows that the presumed counterexamples were using common derivative formulas and were extending them to infinity, even in cases where those common formulas do not represent derivatives there anymore. An ultimate representative result of the new analysis is that if a function tends to a finite limit as time tends to infinity, its derivative must also tend to a zero limit, no matter how the function behaves for finite argument values.

As new results show, all that is needed from the nonlinear system under analysis is the guarantee that bounded trajectories cannot pass an infinite distance in finite time. As the references show, if this condition is satisfied (and it is usually satisfied), then the differential equation may contain discontinuities and even Dirac impulses.

We reproduce here the new theorem of stability, as formulated by Barkana [24], for the general nonlinear non-autonomous system

$$\dot{\mathbf{x}}(t) = \mathbf{f}(\mathbf{x}, t) \tag{2}$$

because it greatly simplifies the stability analysis and also ends up being more conclusive about the ultimate behavior of system trajectories.

The new theorem is based on a simple assumption.

**Assumption 1.** $\int_{\alpha}^{\beta} \|f(x(\tau), \tau)\| d\tau$ *is bounded along any* bounded *trajectory* $\mathbf{x}(t)$ *and for any* finite *time interval* $p = \beta - \alpha$.

Of course, this is only an assumption that must be checked along the bounded trajectories of the system. However, as it only implies that bounded trajectories cannot pass an infinite distance in finite time, it totally eliminates the need to even mention continuity in the context of stability, and it is satisfied in most cases. The system function $\mathbf{f}(\mathbf{x}, t)$ can even contain impulses and even an infinite impulse sequence, and the only "limitation" is that it should not contain an infinite number of impulses in any finite time interval.

In order to formulate the new theorem of stability in its most general form, let us assume that the Lyapunov derivative has the form

$$\dot{V}(\mathbf{x}, t) = W_1(\mathbf{x}, t) + W_2(\mathbf{x}, t). \tag{3}$$

The first term is negative semidefinite, $W_1(\mathbf{x}, t) \leqslant 0$, while the second term, $W_2(\mathbf{x}, t)$, is a transient, not necessarily negative term, which is bounded for bounded $\mathbf{x}$ and $\lim_{t \to \infty} (W_2(\mathbf{x}, t)) = 0$.

We now define the domain $\Omega_i$ as

$$\Omega_i = \left\{ \mathbf{x} \,\big|\, \lim_{t \to \infty} (W_1(\mathbf{x}, t)) \equiv 0 \right\}. \tag{4}$$

Now, we can write the new theorem of stability in the following simple formulation:

**Theorem 1.** *(The new Theorem of Stability) Let $V(\mathbf{x})$ be a one-to-one differentiable function bounded from below. Assume that its derivative $\dot{V}(\mathbf{x}, t)$ along the trajectories of (2) satisfies the conditions defined in (3). Then, if Assumption 1 is satisfied, all limit points of any bounded trajectory $\mathbf{x}(t)$ belong to the domain $\Omega_i$, cf. [24].*

**Remark 1.** *Note that, in order to cover more general nonlinear systems, $V(\mathbf{x})$ is not required to be positive definite. Furthermore, it only has to be differentiable in the sense of Dini. In other words, as long as the Lyapunov derivative satisfies condition (3), it does not have to be continuous or bounded, and it can contain $\delta$-functions or other unbounded functions.*

It is worth emphasizing again that, besides the mere existence of solutions, the *only* condition that the new Theorem of Stability 1 requires is that bounded trajectories cannot pass an infinite distance in finite time.

Moreover, and maybe even more importantly, it is worth mentioning that *all* previous methodologies, including LaSalle's, end with the same conclusion, namely, that "$\dot{V}$ tends to zero as time tends to infinity." Therefore, the customary conclusion for the stability analysis of adaptive control methodologies has been "we don't know what the ultimate behavior of the adaptive gains is, yet at least we know that the following error vanishes as time tends to infinity."

However, according to early LaSalle's own observation and, much more so, to Matrosov's school approach [27], this does not seem to be necessarily true. Actually, as they observed, the mere conclusion "$\dot{V}$ tends to zero as time tends to infinity" does not guarantee much because $\dot{V}$ may reach zero only to leave it and go through other, nonzero, values in order to come back and leave again and then keep coming and leaving forever.

Although the system trajectories could still end at the origin or at other equilibrium points or along some limit cycle, there is no way to know it from the mere result "$\dot{V}$ tends to zero as time tends to infinity."

As this result actually does not promise more than mere stability, which already was there in Lyapunov's original Theorem, Matrosov's school [27] suggests devising and using a few Lyapunov functions and their derivatives for the same system in an attempt to reach more satisfactory conclusions regarding the ultimate behavior of trajectories.

Instead, the new theorem of stability 1 simply states that if the derivative of the Lyapunov function is negative semidefinite (and an eventual temporary non-negative additional term is also allowed), all bounded trajectories end within the domain defined by $\dot{V}(t) \equiv 0$ (note: *identically* equal zero, rather than simply equal zero). Therefore, according to the new theorem of stability 1, all state variables and adaptive gains are bounded, and the system ultimately ends within the domain defined by $\dot{V}(t) \equiv 0$. In the adaptive control case, although $\dot{V}(t)$ is only negative semidefinite as it does not contain the adaptive gains, it still is negative definite in $\mathbf{x}(t)$, and the new theorem of stability 1 shows that the system ends with $\mathbf{x}(t) \equiv 0$. In other words, $\mathbf{x}(t)$ not only ultimately reaches zero but also *stays zero* thereafter, and the adaptive control system indeed demonstrates asymptotic convergence of the state and boundedness of the adaptive gains. Moreover, $\dot{V}(t) \equiv 0$ not only implies that $\dot{V}(t)$ ends up being zero but also that all its consecutive derivatives end up being zero, and this provides sufficient relations to determine the ultimate behavior of all state variables, even if they do not show in $\dot{V}(t)$.

## 2.2. Speed-Gradient Method

At the end of 1970s, it turned out to be possible that a unification of various control algorithms could be achieved if the gradient of the rate of objective function change along the trajectories of the controlled plant was employed; see [13]. The resulting algorithms were called Speed-Gradient (SG) algorithms.

A fairly complete exposition of the SG method can be found in [13–17]. This method is employed in this paper for the nonlinear robust control law design.

Let, for plant model (1), it be required to obtain "admissible control laws" (algorithms) having the form

$$\theta(t) = \Theta\left(\left\{x(s)_{s=0}^t\right\}, \left\{\theta(s)_{s=0}^t\right\}\right) \tag{5}$$

with some operator $\Theta$ such that the solutions of the system (1), (5) exist and are unique for $t \geqslant 0$ for any initial values $x_0, \theta_0$.

When synthesizing the algorithm, a control goal is considered given, expressed by the asymptotic relation

$$\lim_{t\to\infty} Q_t = 0. \tag{6}$$

or inequalities

$$Q_t \leqslant \Delta \ \forall \ t \geqslant t_*, \tag{7}$$

where $Q_t = Q\left(\left\{x(s)_{s=0}^t\right\}, \left\{\theta(s)_{s=0}^t\right\}\right)$ is a given objective functional, $t_* < \infty$.

Two types of functionals are considered [16]:

$$\text{1. } \textit{Local objective functional } Q_t = Q(x(t), t), \ Q(\cdot) \in \mathbb{R}, \tag{8}$$

$$\text{2. } \textit{Integral objective functional } Q_t = \int_0^t q(x(s), \theta(s), s)\,ds. \tag{9}$$

In specific tasks, the control goal may contain some additional conditions. For example, for the integral objective functional, an additional objective goal is also used

$$\lim_{t\to\infty} q(x(s), \theta(s), s) = 0. \tag{10}$$

The SG algorithm in its "differential form" is the vector $\theta$ law of variation, given by the following differential equation, cf. [13]:

$$\frac{d\theta}{dt} = -\Gamma\nabla_\theta\omega(x, \theta, t), \quad \theta(0) = \theta_0, \ t \geqslant 0, \tag{11}$$

where $\omega(x, \theta, t)$ is the derivative of the objective functional due to the system (1); $\Gamma = \Gamma^{\mathrm{T}} > 0$ is a positive-definite $m \times m$-matrix of algorithm gains. The $\Gamma$ matrix is often chosen to be diagonal ($\Gamma = \mathrm{diag}\{\gamma_1, \ldots, \gamma_m\}$) or scalar ($\Gamma = \gamma\mathbf{I}_m$). Under sufficiently general conditions, algorithm (11) ensures that system (1) achieves the goal (6). The exact formulation is given in [13]. Further development of the SG method led to the appearance of more general algorithms [14–16], in which the derivative of the anti-gradient of the rate of change in the objective functional is introduced into control law (11): $\frac{d\theta}{dt} = -\Gamma_2\nabla_\theta\omega(x, \theta, t) - \Gamma_1\frac{d}{dt}\nabla_\theta\omega(x, \theta, t)$, where $\Gamma_1, \Gamma_2$ are symmetric positive-definite gain matrices. Algorithms of such a structure correspond to the "classical" Proportional Differential (PD) controllers if $\theta(t)$ is taken as the control signal $u(t)$.

This approach leads to the following algorithm:

$$\theta(t) = -\Gamma \int_0^t \nabla_\theta \omega(x, \theta, \tau)\, d\tau - \psi(x, \theta, t) + \theta_0, \tag{12}$$

where $\Gamma = \Gamma^{\mathrm{T}} > 0$ stands for $m \times m$-matrix of algorithm gains; $\omega(x, \theta, t)$ denotes the derivative of the objective functional along system (1) trajectories; $\psi(x, \theta, t)$ is a certain vector-function satisfying the following "pseudogradient condition":

$$\psi(x, \theta, t)^{\mathrm{T}} \nabla_\theta \omega(x, \theta, t) \geqslant 0. \tag{13}$$

For example, as $\psi(x, \theta, t)$ one can take

$$\psi(x, \theta, t) = \Gamma_1 \nabla_\theta \omega(x, \theta, t), \tag{14}$$
$$\psi(x, \theta, t) = \Gamma_2 \operatorname{sign}\left(\nabla_\theta \omega(x, \theta, t)\right), \tag{15}$$

where $\Gamma_i = \Gamma_i^{\mathrm{T}} > 0 - m \times m$-matrices ($i = 1, 2$), and $\Gamma_2$ is diagonal.

*2.3. SG Algorithms in Finite-Differential Form*

Consider algorithm (12)

$$\theta(t) = -\Gamma \int_0^t \nabla_\theta \omega(x, \theta, \tau)\, d\tau - \psi(x, \theta, t) + \theta_0, \tag{16}$$

where, as above, $\omega(x, \theta, t)$ is the derivative of the objective functional due to the system (1); $\psi(x, \theta, t)$ is a vector function satisfying the condition (13). Let us extend the class of algorithms (12), assuming that instead of the condition of positive definiteness of the gain matrix $\Gamma$, a more general condition of non-negative definiteness is satisfied: $\Gamma = \Gamma^{\mathrm{T}} \geqslant 0$. Thus, the degeneracy of this matrix is allowed. As shown later in the paper, this allows one to obtain a wider class of speed gradient algorithms in the finite-differential form. In such combined algorithms, the parametric adjustment of the controller is joined with the introduction of a signal component into the control law.

The following condition for the applicability of algorithm (16) with a degenerate matrix $\Gamma$ and a local objective functional (8) is derived.

**Theorem 2** ([14]). *Let:*

*— For all $v \in \mathbb{R}^m$, there is a unique solution $\theta = \kappa(x, v, t)$ to equations $\theta + \psi(x, \theta, t) = v$,*

*— Functions $f(x, \theta, t)$, $\nabla_x Q(x, t)$, $\psi(x, \theta, t)$, $\nabla_\theta \omega(x, \theta, t)$ bounded in any bounded set $\{\|x\| + \|\theta\| \leqslant \beta,\ t \geqslant 0\}$;*

*— The growth condition $\inf\limits_{t \geqslant 0} Q(x, t) \to \infty$ as $\|x\| \to \infty$;*

*— Function $\omega(x, \theta, t)$ is convex in $\theta$;*

*— There exists a vector $\theta_* \in \mathbb{R}^m$ and a function $\rho(Q)$ ($\rho(Q) > 0$ for $Q > 0$) such that for all $x$, $t$ is valid:*

$$\omega(x, \theta_*, t) \leqslant -\rho(Q). \tag{17}$$

*is valid. Then, all the system trajectories with initial conditions belonging to set $\Omega_0 = \{(x, \theta) : (\mathbf{I}_m - \Gamma^\dagger \Gamma)(\theta_0 - \theta_*) = 0\}$ are bounded and $\lim\limits_{t \to \infty} Q_t = 0$, i.e., the control goal (6), is achieved. (here $\Gamma^\dagger$ denotes a matrix that is pseudo-inverse to matrix $\Gamma$.)*

Unlike [14], in the present paper, Theorem 2 is proved without employing the Barbalat Lemma but is based on the new theorem of stability 1, which eliminates the apparently necessary continuity conditions.

**Proof.** Introduce the Lyapunov function

$$V(x,\theta,t) = Q(x,t) + \tfrac{1}{2}\|\theta - \theta_* + \psi(x,\theta,t)\|_{\Gamma^\dagger\Gamma}^2 \tag{18}$$

Calculating its time derivative due to the system (1) and (16), we obtain

$$\dot{V}_t = \omega(x(t),\theta(t),t) + v_t^\mathrm{T}\Gamma\nabla_\theta\omega(x(t),\theta(t),t), \tag{19}$$

where $\omega(x,\theta,t)$ is given by (16), $v_t = \theta(t) - \theta_* + \psi(x(t),\theta(t),t)$. According to the condition, $v_0 \in \mathbb{L}(\Gamma)$, where $\mathbb{L}(\Gamma)$ is the linear span columns of matrix $\Gamma$. Algorithm (16), $\frac{dv_t}{dt} \in \mathbb{L}(\Gamma)$. Consequently, $v_0 \in \mathbb{L}(\Gamma)$ for all $t \geqslant 0$, so that $\Gamma^\dagger\Gamma v_t = v_t$ ($\Gamma^\dagger\Gamma$ is projector onto the set $\mathbb{L}(\Gamma)$). Therefore, (19) becomes $\dot{V}_t = \omega(x(t),\theta(t),t) + v_t^\mathrm{T}\nabla_\theta\omega(x(t),\theta(t),t)$. Applying now the convexity conditions and reachability, we obtain $\dot{V}_t \leqslant -\rho(Q(x(t),t)) \leqslant 0$. Hence $V(x(t),\theta(t),t) \leqslant V(x_0,\theta_0,0)$, which proves the boundedness of the trajectories of system (1) and (16). So $\int_0^\infty \rho(Q(x(t),t))dt < \infty$, whence using the new theorem of stability 1, one deduces that $\lim_{t\to\infty} Q(x(t),t) = 0$. This completes the proof. $\square$

An important special case of (16) is the finite form of the SG algorithm, which can be written as (16) for $\Gamma = 0$:

$$\theta = \theta_0 - \gamma\psi(x,\theta,t), \tag{20}$$

where $\gamma > 0$ is the algorithm parameter (the "gain factor").

The SG method extension to time-varying nonlinear plants is presented in [28].

*2.4. Combined Algorithms for Adaptive Control with Implicit Reference Model*

Following [15–17], let us introduce the following signal-parametric adaptation algorithms in systems with the Implicit Reference Model (IRM).

Consider a controlled plant whose output signal directly represents the "error" (discrepancy) $\sigma(t)$ as follows;

$$\dot{x}(t) = Ax(t) + Bu(t), \quad \sigma(t) = g^\mathrm{T}x(t), \tag{21}$$

where $x(t) \in \mathbb{R}^n$, $u(t) \in \mathbb{R}^m$, $\sigma(t) \in \mathbb{R}^1$

Let us pose the control goal in the form of the limit relation $\lim_{t\to\infty} x(t) = 0$, and following the SG design method, let us pick up the objective function as $Q_t = \frac{1}{2}\sigma(t)^2$ and introduce the auxiliary control goal as ensuring the finite-time convergence of $Q_t$ to zero: $Q_t = 0$ for all $t \geqslant t_*$, where $0 < t_* < \infty$. Note that such an auxiliary goal is also typical for systems with the variable structure on sliding modes [29–36].

Following the SG scheme, one obtains

$$\dot{Q}_t = \omega(x,\theta,t) = g^\mathrm{T}x(g^\mathrm{T}Ax + g^\mathrm{T}Bu(t)). \tag{22}$$

Let the control law in the main feedback loop be taken as

$$u(t) = K(t)x(t) + u_s(t), \tag{23}$$

where $\theta(t) = \mathrm{col}\{K(t), u_s(t)\}$ denotes the vector of adjustable parameters.

Then, one obtains

$$\begin{aligned}\nabla_K\omega(x,\theta,t) &= (g^\mathrm{T}B)g^\mathrm{T}xx^\mathrm{T}, \\ \nabla_{u_s}\omega(x,\theta,t) &= (g^\mathrm{T}B)g^\mathrm{T}x.\end{aligned} \tag{24}$$

Then the SG algorithm of adaptive control in finite form (20) takes the form

$$u(t) = K(t)x(t) - \gamma \text{sign}(g^{\text{T}}B)\sigma(t),$$
$$K(t) = -\gamma_1(g^{\text{T}}B)\sigma(t)x(t)^{\text{T}}. \tag{25}$$

Let us note that in the system (21), (25) a sliding mode occurs on the surface $g^{\text{T}}x = 0$; therefore, to achieve the control goal $\lim\limits_{t \to \infty} g^{\text{T}}x(t) = 0$, transfer function $W(s) = g^{\text{T}}(s\mathbf{I} - A)^{-1}B$ from control input $u$ to measured output $\sigma$ should be strictly minimal phase [15,16,37].

The control algorithm with the IRM using only output $\sigma(t)$ measurements can be derived as the SG algorithm in the finite form (20), which in the considered case takes the form of a relay control law:

$$u(t) = -\gamma \text{sign}(g^{\text{T}}B)\sigma(t), \quad \sigma(y) = g^{\text{T}}y, \quad \gamma > 0. \tag{26}$$

The algorithm can be considered as a particular case of (25), where $K(t) \equiv 0$.

In the present paper, this form of the control law is used for robust control of the phase shift between the revolving rotors of the two-rotor vibration machine.

## 3. Experimental Setup: Two-Rotor Mechatronic Vibration Machine
### 3.1. Setup Description

The vibration complex used for experiments in this work has ample opportunities for research. The complex can be used to study such problems of dynamics as vibrational rotation, braking and starting of an unbalanced rotor, the Sommerfeld effect, self-synchronization of vibration exciters, synchronization control, and vibration isolation from disturbances. This complex, see Figure 1, consists of a vibration stand with electric drives of unbalanced rotors, an electronic unit of an amplifier converter for powering electric motors, a sensor system with a special signal processing controller for measuring the parameters of the stand, a personal computer with interface devices with physical equipment, software for real-time control algorithms and processing of received information; see [38]. The setup is described in many details in [9–11]; therefore, in the present paper, only a brief exposition is given. All devices of the complex are interconnected in a single closed system in which mechanical processes and control processes take place.

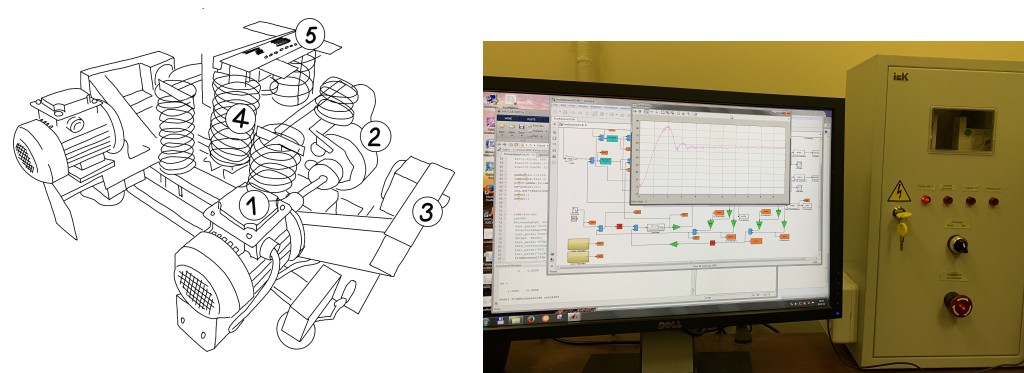

1 – AC motors, 2 – vibration actuators, 3 – support frame, 4 – helical springs, 5 – vibrating platform

**Figure 1.** Axonometric view of mechanical part (**left**) and control board photo (**right**) of laboratory mechatronic setup MMLS SV-2M.

The basis of the mechanical part of the complex is a pair of unbalanced vibration exciters, each of which consists of three-phase induction motors (IM) (1) with a controlled speed, rigidly connected through a shaft with vibration actuators (2), which are unbalanced rotors, revolving on the shaft in a vertical plane on the stand body at 2760 rpm, and the rated electromagnetic torque is 0.04 Nm. The operating speed range of the rotors is from 20 to 120 rad/s. The imbalance of the rotor is achieved by an eccentrically located load and can be installed in three positions. Anti-vibration helical springs (4) are used to reduce

the transmission of vibrations of the bench table to support frame (3) and the base under it. The working platform is attached to base 5. The complex is powered by a single-phase alternating current with a frequency of 50 Hz and a voltage of 220 V and is turned on through the electronics-computer facility (Figure 1, right).

Figure 2 shows the functional electrical diagram of the setup. Power frequency converters (FC) are used to control right (RM) and left (LM) electric motors. The FCs are connected to the computer via the RS-485 interface and analog control.

The stand is equipped with a system of sensors for eight linear and angular displacements of the platform. Each platform has three linear optical sensors and three optical angular displacement sensors.

### 3.2. Servo System Model

The mechanical part of setup SV-2M, pictured in Figure 1, contains two oscillatory platforms with two groups of springs and has seven degrees of freedom (DoF). Its kinematics is represented in Figure 2 of [39]. The corresponding dynamics model can be derived from Equations (1) and (2) of [39], based on the standard Lagrangian formalism. However, in [39], it is assumed that the control actions are input torques. However, the torques cannot be directly controlled by the outer equipment or even measured for the existing setup. These torques are applied from the pair of computer-controlled Induction Motors (IM). The dynamics of MI are known to be very complex; the complexity of the overall dynamics of the system also increases dramatically due to the cross-coupling of the mechanical part and the electrodynamics of the IM. Moreover, the influence of platform vibrations on the rotation of unbalanced rotors also leads to the mutual cross-coupling of the rotors' movement, which for the case of pendulum systems was described by Christian Huygens [40]. For multi-rotor machines, this phenomenon is known as self-synchronization of the rotors and was observed by I.I. Blekhman as the revolving of the electric drive rotor, which is not switched on to the power circuit. The interaction of rotating rotors at sufficiently high frequencies leads to the phenomenon of self-synchronization, which is difficult (or even impossible) to overcome by changing the electrical control signal. The rotor self-synchronization phenomena were deeply investigated in [1,7,10,41–45]. It also should be taken into account that, in the system, the industrial frequency converters (Altivar 12 Schneider Electric) are used with their own local controllers, the algorithm for which is not revealed by the manufacturer. The Coulomb friction, which inevitably exists in the drives and mechanical transmission of the system, also leads to effects that are difficult to model accurately, cf. [46–48].

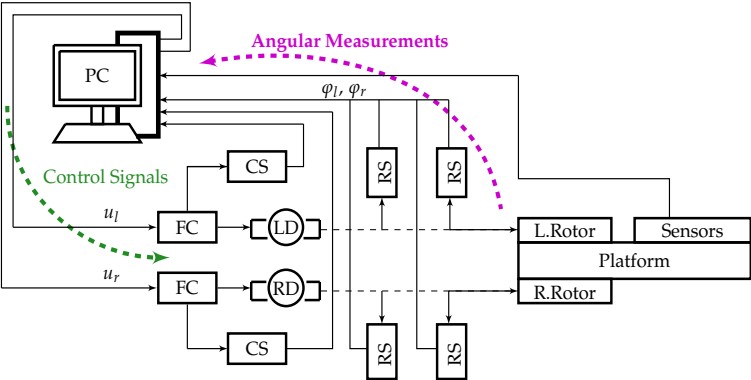

**Figure 2.** Functional electrical diagram for setup control.

Although the dynamics of the system are very complex, since we are interested here in the revolving of the rotors and not the vibrations of the platform, it is advisable to use a simplified model for certain conditions assumed for this study that reflects the motor gains and main time constants with suitable accuracy, using the [10] approach. For SV-2M, a significant influence of the gravitational (" pendulum") torque of the engine is

observed at low frequencies (up to 5 Hz), which cannot be ignored for the design of the regulator (cf. [39,49]. However, in the middle and high-frequency ranges (5–20 Hz), the so-called "averaging property", i.e., when fast oscillatory components are averaged, and for rotating rotors, only "slow" movements can be taken into account; see [1]. In addition, since the induction motors have their local feedback controllers, the dynamics of the drive systems, including the induction motor and the frequency converter with the feedback local controller, can be approximately described by the second-order transfer function from the control signal to angular velocity $\omega$, cf. [50–52] as follows:

$$W_d(s) = \left\{ \frac{\omega}{u} \right\} = \frac{b_0}{a_0 s^2 + a_1 s + 1} = \frac{k_d}{T^2 s^2 + 2\xi T s + 1}, \tag{27}$$

where $b_0$, $a_0$, $a_1$ stand for the drive model parameters, where $b_0 = k_d$ corresponds to the drive system static gain; $T = \sqrt{a_0}$ is the time constant; $\xi = a_1(2T)^{-1}$ denotes the damping ratio; $s \in \mathbb{C}$ stands for the Laplace transform variable. Note that the case of $\xi \leqslant 1$ is also possible. Then the denominator of (27) can be rewritten as $T^2 s^2 + 2\xi T s + 1 = (Ts + 1)(\tau s + 1)$, where $T_1 \tau = T^2$, $T_1 + \tau = 2\xi T$. Model (27) is used in this work at the stage of the controller design and for the preparatory simulations before providing the real-world experiments in the SV-2M setup.

### 3.3. Identification of Motor Model Parameters

The standard non-recursive least squares (LSE) method was used to identify the parameters of the drive model, cf. [53]. To achieve this, in [10], the drive systems were excited to obtain data for the identification procedure by applying offset input signals of a rectangular shape $u_l$, $u_r$, providing a change in the angular velocities of the drive systems up to about 10 rad/s around the "base" speed $\omega_0$. The excitation waveform period of 10 s and value $\omega_0 = 60$ rad/s were set.

To this end, the sampled-data version of model (27) is transformed into the linear regression form

$$y[k] = \zeta[k]^{\mathsf{T}} \theta + v[k], \tag{28}$$

see [53] for details.

For the considered case, model (27) is represented as

$$\omega(t) = \theta_1 u(t) - \theta_2 \ddot{\omega}(t) - \theta_3 \dot{\omega}(t), \tag{29}$$

where $\theta_1 = b_0$, $\theta_2 = a_0$, $\theta_3 = a_1$. Since the angular velocity $\omega$ is not explicitly measured, at the stage of parameter estimation, it is replaced by its zero-hold approximation $\tilde{\omega}(t) = \tilde{\omega}[k]$ for $t \in [t_k, t_{k+1}]$, where $t_k = kT_0$, $T_0$ is the sampling interval, $k = 0, 1, \ldots$, and $\tilde{\omega}[k]$ is an output of the finite impulse response (FIR) digital differentiator $\tilde{\omega}[k] = (\varphi[k] - \varphi[k-1])/T_0$, $\varphi[k] = \varphi(t_k)$ is a measured value of $\varphi(t)$ at instant $t_k$. The resulting noise caused by quantization of $\varphi(t)$ by the optical sensors is then suppressed by the Least-Square Estimation (LSE) procedure. Three third-order low-pass filters are introduced as

$$W_f(s) = \frac{\mu^3}{s^3 + 3\mu s^2 + 3\mu^2 s + \mu^3}, \tag{30}$$

where $\mu > 0$ stands for the filter bandwidth. Signals $\tilde{\omega}(t)$, $u(t)$ are fed to the corresponding filters (30). Finally, in (28), one has $m = 3$, $\zeta \in \mathbb{R}^3$, $y[k] = \tilde{\omega}_f(t_k)$, $\zeta_1[k] = -u_f(t_k)$, $\zeta_2[k] = \ddot{\omega}_f(t_k)$, $\zeta_3[k] = \dot{\omega}_f(t_k)$, where subscript $f$ is referred to the corresponding output of the state filters, cf. [53–55]. This procedure leads to the estimate $\hat{\theta}$ for $\theta$ as $\hat{\theta} = \Phi^{\dagger} Y$, where

$$\Phi = \begin{bmatrix} \zeta_1[1] & \zeta_2[1] & \zeta_3[1] \\ \vdots & \vdots & \vdots \\ \zeta_1[N] & \zeta_2[N] & \zeta_m[N] \end{bmatrix}, \quad Y = \begin{bmatrix} y[1] \\ \vdots \\ y[N] \end{bmatrix}. \tag{31}$$

In (31), $\Phi^\dagger$ denotes the pseudo-inverse matrix to $\Phi$. The pseudo-inverse operation is made by the standard MATLAB routine *pinv*.

## 4. Synchronization and Phase Shift Control of Vibration Setup Actuators

In [9], the following "bidirectional" control law was proposed:

$$e_{\omega_l} = \omega_l^* - \omega_l, \quad e_{\omega_r} = \omega_r^* - \omega_r, \tag{32}$$

$$\dot{\delta}_{\omega l} = e_{\omega_l}, \quad u_{\omega_l} = K_{i\omega_l}\delta_{\omega l} + K_{p\omega_l}e_{\omega_l}, \tag{33}$$

$$\dot{\delta}_{\omega r} = e_{\omega_r}, \quad u_{\omega_r} = K_{i\omega_r}\delta_{\omega r} + K_p e_{\omega_r}, \tag{34}$$

$$\psi = \varphi_r - \varphi_l, \tag{35}$$

$$e_\psi = \psi^* - \psi, \tag{36}$$

$$\dot{\sigma}_\psi = \sin e_\psi, \quad u_\psi = -K_{i,\psi}\sigma_\psi + K_{p,\psi}\sin e_\psi, \tag{37}$$

$$u_l = \text{sat}_0^{u_{\max}}(u_{\omega_l} + u_\psi), \quad u_r = \text{sat}_0^{u_{\max}}(u_{\omega_r} - u_\psi), \tag{38}$$

where $\omega_r^*$, $\omega_l^*$; $e_{\omega_l}$, $e_{\omega_r}$ are the engine speed errors; the PI speed controllers of the left and right motors are described, respectively; see [9] for more detail. Function $\text{sat}_0^{u_{\max}}(\cdot)$ in (38) corresponds to the natural saturation of the FC inputs: $u_l$, $u_r$ are non-negative and should not exceed some maximum value $u_{\max}$. Due to hardware features $u_{\max} = 2^{16} - 1$, but out of caution in this work, $u_{\max} = 40000$ is set. Controller (32)–(38) consists of three PI controllers with symmetrical cross-coupling between them. The phase shift control law (35)–(37), as a part of the common control Algorithm (32)–(38), is used to ensure the prescribed phase shift $\psi^*$ between the rotors.

The adaptive sampled-data variant of the control law (32)–(38) is proposed and studied in [11] for the multiply-synchronization case. In the present study, the phase shift control algorithm (32)–(38) (and its adaptive variant of [11]) is replaced by the non-tunable robust controller, following from (26). Namely, to avoid the steady state error and, at the same time, keep the level $\gamma$ of the relay term in (26) as small as possible, the integrator is added to the plant input, controlled by another relay component of the control action.

In summary, instead of (32)–(38), the following control law is proposed:

$$e_{\omega_l} = \omega_l^* - \omega_l, \quad e_{\omega_r} = \omega_r^* - \omega_r, \tag{39}$$

$$\dot{\delta}_{\omega l} = e_{\omega_l}, \quad u_{\omega_l} = K_{i\omega_l}\delta_{\omega l} + K_{p\omega_l}e_{\omega_l}, \tag{40}$$

$$\dot{\delta}_{\omega r} = e_{\omega_r}, \quad u_{\omega_r} = K_{i\omega_r}\delta_{\omega r} + K_p e_{\omega_r}, \tag{41}$$

$$\psi = \varphi_r - \varphi_l, \tag{42}$$

$$e_\psi = \psi^* - \psi, \tag{43}$$

$$\Delta\omega = \omega_l - \omega_r, \tag{44}$$

$$\sigma = e_\psi + \tau_M \Delta\omega, \tag{45}$$

$$\dot{v} = \gamma_I \,\text{sign}(\sigma), \tag{46}$$

$$u_\psi = \text{sat}_{\bar{u}_\psi}\left(v + \gamma\,\text{sign}(\sigma)\sqrt{|\sigma|}\right), \tag{47}$$

$$u_l = \text{sat}_0^{u_{\max}}(u_{\omega_l} + u_\psi), \quad u_r = \text{sat}_0^{u_{\max}}(u_{\omega_r} - u_\psi), \tag{48}$$

The "square-root" term in (47) is inspired by the super-twisting method of Levant [30], Bartolini et al. [35]. By the analogy with (37), also the "sine-modification" can be used, where (45) is replaced by

$$\sigma = \sin e_\psi + \tau_M \sin(\Delta\omega). \tag{49}$$

This modification is based on the fact that from the point of view of the phase mismatch of the rotors for vibration systems, the difference in the total number of revolutions does not matter, and at the same time, at the initial stage of rotors revolving, they do not move synchronously; therefore, a significant phase shift accumulates, which can lead to large values of the control signal $u_\psi$. This circumstance turned out to be significant for PI controllers in the phase control loop, especially in their adaptive version. Comparative analysis based on experimental data (see Section 5) showed, however, that this effect is of minor importance in the proposed relay control, and therefore a "linear" discrepancy signal $\sigma(t)$ in the form of (45) can be used.

The algorithm represented by equations belongs to algorithms of the IRM type (with an implicit reference model, see). In the case under consideration, this model is expressed by the identity $\sigma(t) \equiv 0$, and the applied control law, according to the SG method, ensures the decrease in $|\sigma(t)|$ (see Section 2.2, Equation (10)). Since signal $\sigma$ of the form (45) can be rewritten as $\sigma = e_\psi + \tau_M \Delta\omega = \psi^* - \psi + \tau_M(\omega_l - \omega_r) = \psi^* - \psi - \tau_M\dot\psi$, then equivalence $\sigma(t) \equiv 0$ means fulfillment of equation $\tau_M\dot\psi + \psi = \psi^*$, which can be called "the reference equation" by the analogy with the habitual reference model in Model Reference Adaptive Control (MRAC), cf. [56]. The difference to the MRAC approach is that the reference model

$$\tau_M\dot\psi_M(t) + \psi_M(t) = \psi^*(t) \tag{50}$$

is not a part of the system but implicitly represented by parameter $\tau_M$ of the algorithm. Based on this property, this method is called the "IRM method". Although signal $\psi_M$ is not used in the control law (32)–(48), it is shown, for clarity, together with $\psi(t)$ time histories in the experimental part; see Section 5.

**Remark 2.** *As we already wrote, Assumption 1 is satisfied in all cases when bounded trajectories cannot pass an infinite distance in finite time. How could this assumption be violated? Even if the differential equation contains impulses, they would only lead to bounded jumps of the trajectories. Could a sequence of impulse functions make these bounded jumps sum to an infinite distance in finite time? It is easy to see that this could occur only if there is an infinitely dense sequence of impulse functions in a finite interval. As this is improbable in any realistic plant, Assumption 1 is usually satisfied.*

## 5. Numerical Examination and Experimental Results

In this section, the simulation and experimental results for the phase shift control are presented. All the experiments have been carried out on the mechatronic vibrational setup MMLS SV-2M, as described in Section 3. The desired rotation frequency $\omega^* = 60$ rad/s was set to both rotors. The PI-controller gains for rotational frequency were taken as $k_I = 240$ s, $k_P = 1680$, both for right and left motor loops. Control law (39)–(49) parameters were taken as $\gamma = 1000$, $\gamma_I = 500$. The controller sampling interval $T_s$ was 0.02 s. For the simulations, applying the LSE identification procedure of Section 3.3, the following parameters of the left and right drive system models (27) were found: $b_0 = 0.0042$ s$^{-1}$, $a_0 = 0.119$ s$^2$, $a_1 = 0.811$ s, which gives $k_{d_l} = 0.0042$ s$^{-1}$, $T_l = 0.62$ s, $\tau_l = 0.19$ s (for the left drive); $b_0 = 0.0043$ s$^{-1}$, $a_0 = 0.1185$ s$^2$, $a_1 = 1.2195$ s, which gives $k_{d_r} = 0.0043$ s$^{-1}$, $T_l = 1.11$ s, $\tau_l = 0.11$ s (for the right drive). The IRM (50) time constant $\tau_M = 1$ s was taken. In (47), $\bar u_\psi = 5000$.

As results of experiments and simulations, the responses of the system to a constant, harmonic and chaotic reference signal on the phase shift $\psi^*$ between the rotors were obtained.

### 5.1. Case of Constant Reference Phase Shift

The results for the case of the constant reference phase shift $\psi^* \in \{0, \pi/2, \pi\}$ rad are pictured in Figures 3–8.

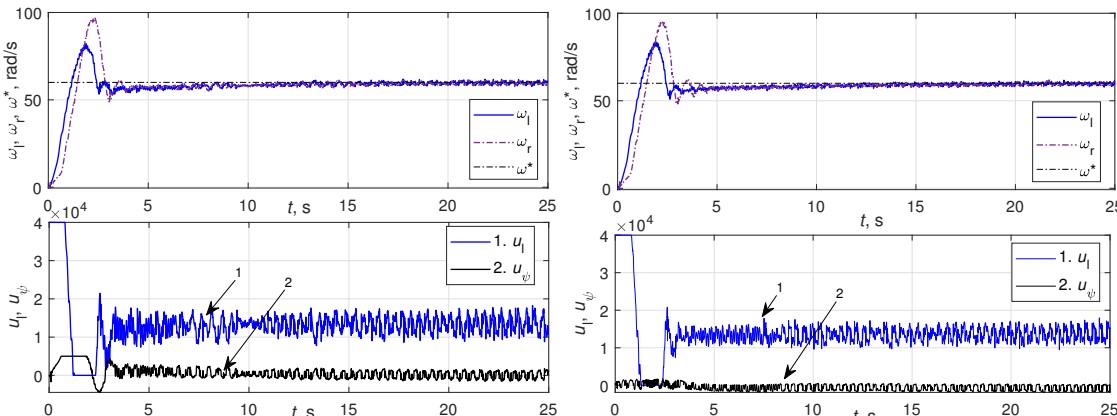

**Figure 3.** Experiment. $\omega_l(t)$, $\omega_r(t)$, $\omega^*(t)$ (**upper plot**) and control actions $u_l(t)$, $u_\psi(t)$ (**lower plot**) time histories for $\psi^* = 0$. Linear error (**left column**), sine-modification (**right column**).

The experimental results for $\psi^* = 0$ are shown in Figures 3–5. It is seen from Figure 3 that after some transient with the settling time approximately equal to 5 s, both the left and the right rotor velocities $\omega_l(t)$, $\omega_r(t)$ are closed to the reference value $\omega^* = 60\ s^{-1}$. The sine-modification of the algorithm with (49) leads to smaller control actions $u_l$ (the control signal for the left drive) and $u_\psi$ (the "phase" control signal) than the usage of the "linear error" (45), but the difference in the control actions' magnitudes is not significant. As is seen from the time histories depicted in Figure 8, the settling time of the phase shift $\psi$ is about 7 s, and the phase shift state error is small for practice. It should be noted that the sinusoidal modification can lead to a deviation of the steady-state phase shift from the specified one by an integer number of revolutions (in this case, by one revolution, angle $2\pi$), which is seen from the phase shift $\psi(t)$ plot in the right column of Figure 4. The overshoot of the phase shift is explained as a result of the difference between the left and right rotors' accelerations due to the different physical properties of the drives. Actually, the regulation problem is considered in this experiment; therefore, the "reference model" output $\psi_M(t) \equiv 0$. This is a case when the customary MRAC systems [56] "go blind".

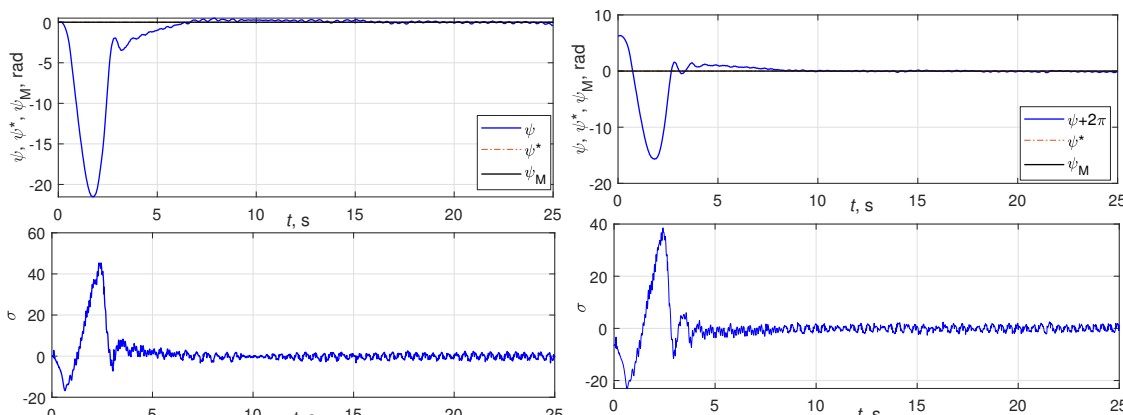

**Figure 4.** Experiment. $\psi(t)$, $\psi^*$, $\psi_M(t)$ (**upper plots**) and IMR discrepancy signal $\sigma(t)$ (**lower plots**) time histories for $\psi^* = 0$. Linear error (**left column**), sine-modification (**right column**).

As stated above, signal $\sigma$, see (26) and (45), can be treated as the discrepancy between the desired and actual system action so that identity $\sigma \equiv 0$ corresponds to the reference model (50) output $\psi_M(t)$. Equation (50), however, is not a part of the control algorithm.

For illustration, the time histories of $\varepsilon_\psi(t) = \psi_M(t) - \psi(t)$ are also plotted for the cases of "linear" discrepancy (45) and the sine-modification (49); see Figure 5.

Similar experimental results for case $\psi^* = \pi$ are demonstrated in Figures 6–8. The plots show that after some transient time of about 10 s, the phase shift $\psi(t)$ reaches the desired value. The settling time for $\sigma(t)$ is about 7 s. The chattering of $\sigma(t)$ is visible, which is caused by the action of platform vibration to rotors motion, time sampling, FIR differentiation, and unmodeled dynamics. This chattering, however, does not make a serious impact on controlled variables $\psi$, $\omega_l$, $\omega_r$ time histories.

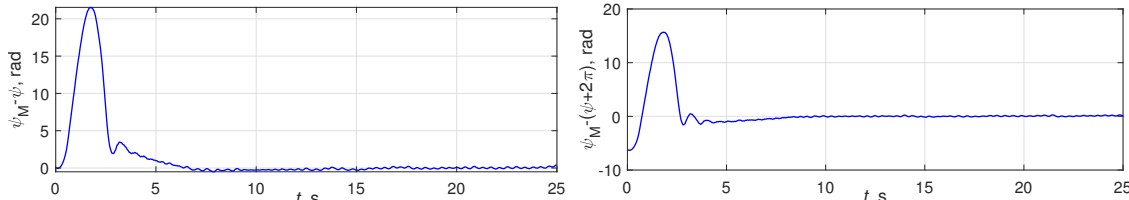

**Figure 5.** Experiment. IMR output error $\varepsilon_\psi(t) = \psi_M(t) - \psi(t)$ time history for $\psi^* = 0$. Linear error (**left plot**), sine-modification (**right plot**).

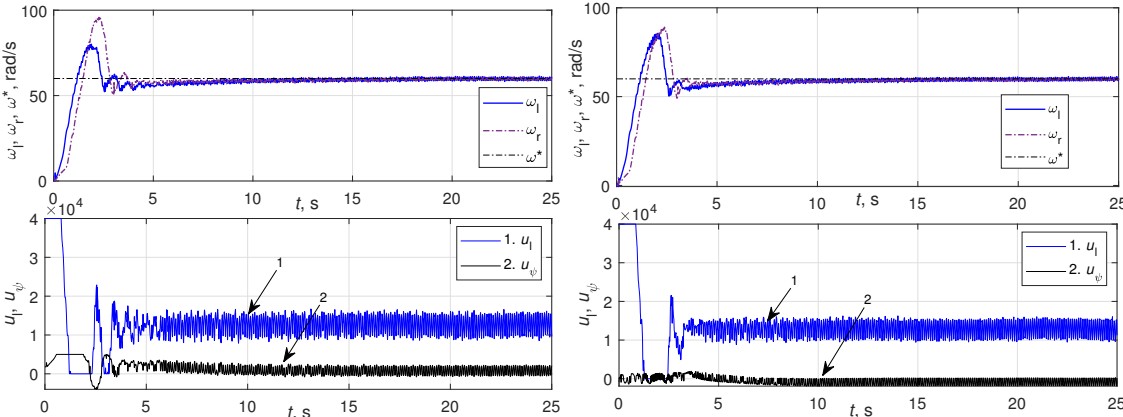

**Figure 6.** Experiment. $\omega_l(t)$, $\omega_r(t)$, $\omega^*(t)$ (**upper plot**) and control actions $u_l(t)$, $u_\psi(t)$ (**lower plot**) time histories for $\psi^* = \pi$. Linear error (**left column**), sine-modification (**right column**).

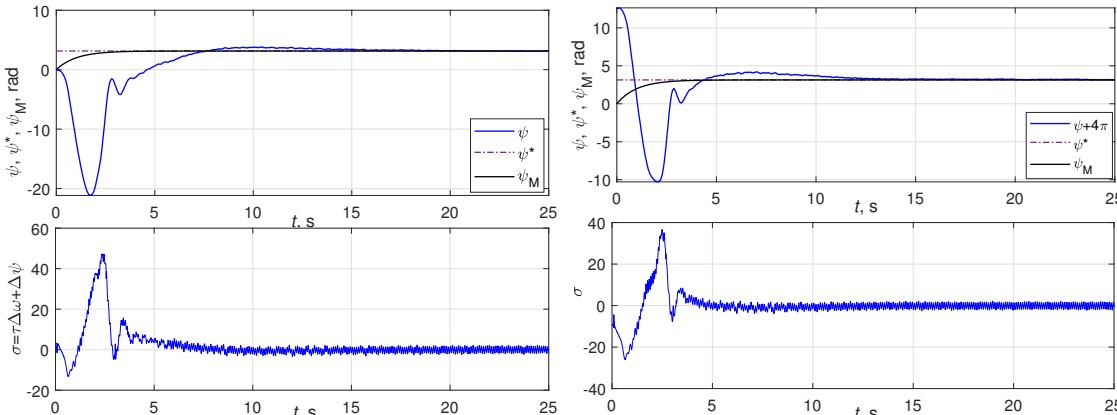

**Figure 7.** Experiment. $\psi(t)$, $\psi^*$, $\psi_M(t)$ (**upper plot**) and IMR discrepancy signal $\sigma(t)$ (**lower plot**) time histories for $\psi^* = \pi$. Linear error (**left column**), sine-modification (**right column**).

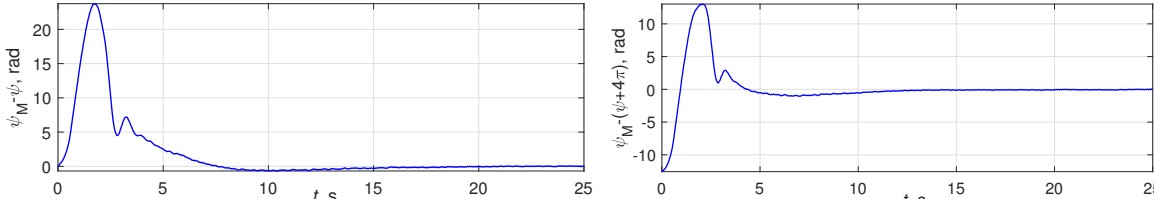

**Figure 8.** Experiment. IMR output error $\varepsilon_\psi(t) = \psi_M(t) - \psi(t)$ time history for $\psi^* = \pi$. Linear error (**left plot**), sine-modification (**right plot**).

## 5.2. Case of Harmonic Reference Phase Shift

The case of harmonic reference signal $\psi^*(t)$ is demonstrated by Figures 9–11. For $\psi^* = \pi \sin(0.1t)$, the simulation and experimental results can be compared by considering Figures 9 and 10. As can be seen from the plots, the processes in the systems have similar characters, but the simulation results, as is usually the case, show better system performance than the experiment. For our system, this phenomenon can be explained, first of all, by the influence of unmodeled setup dynamics and the Coulomb friction between the moving parts. Due to friction, the starting of the right engine is delayed, and as a result, as with a constant reference action, a significant deviation of the rotation speeds of the engines occurs during start-up and, consequently, a large overshoot in phase.

The experimental results for $\psi^* = \pi \sin(0.2t)$, presented in Figure 11, show that after a transient process lasting about 10 s, the system behavior is close to the reference one (with accuracy in 3 revolutions in the case of sin-modification).

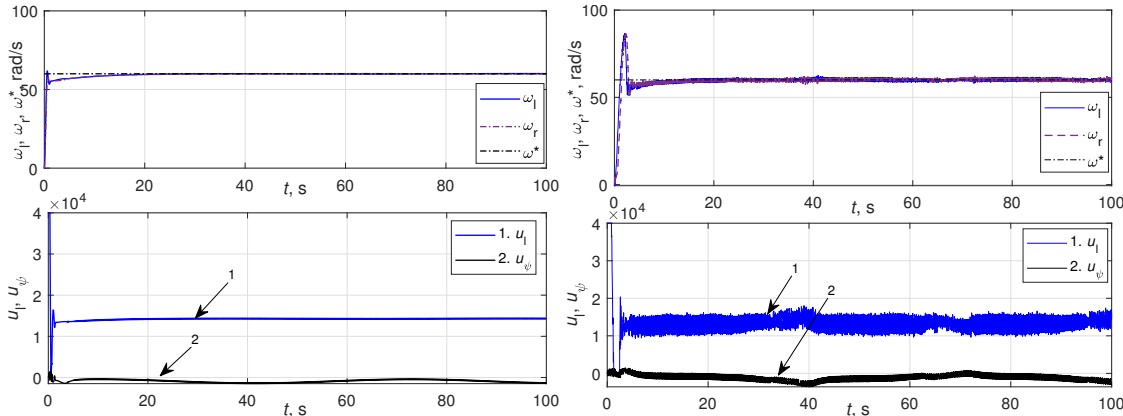

**Figure 9.** Simulation (**left column**) and experiment (**right column**). $\omega_l(t)$, $\omega_r(t)$, $\omega^*(t)$ (**upper plot**) and control actions $u_l(t)$, $u_\psi(t)$ (**lower plot**) time histories for $\psi^* = \pi \sin(0.1t)$. Sine-modification.

## 5.3. Case of Chaotic Reference Phase Shift

Andrievsky and Zaitceva [12] used the Lorentz system as a source of the reference phase shift. The Lorentz generator is represented by (see [57–61]):

$$\begin{cases} \dot{x}_1(t) = m_t\big(28x_3(t) - x_1(t) - x_2(t)x_3(t)\big), \\ \dot{x}_2(t) = m_t\big(x_1(t)x_3(t) - 2.666x_2(t)\big), \\ \dot{x}_3(t) = 10m_t\big(x_1(t) - x_3(t)\big), \end{cases}$$
$$\psi^*(t) = cx_1(t), \tag{51}$$

where $m_t$ and $c$ denote the time and output scaling factors, respectively. The corresponding time histories of $\omega_l(t)$, $\omega_r(t)$, $\omega^*(t)$, $u_l(t)$, $u_\psi(t)$, $\psi(t)$, $\psi^*(t)$, $\psi_M(t)$, $\sigma(t)$ for chaotic $\psi^*(t)$ as an output of system (51) with $m_t = 0.1$, $c = 0.125$ are depicted in Figure 12.

Experimental results with chaotic reference actions, in addition to the direct purpose for the chaotization of the movement of platforms, as in [12], are also helpful for establishing the properties of the system in a wide frequency range, cf. Bucolo et al. [62].

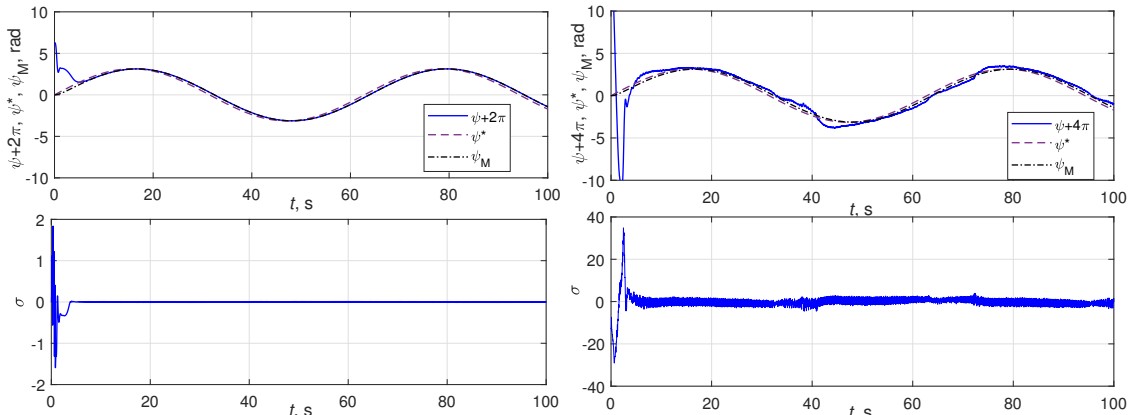

**Figure 10.** Simulation (**left column**) and experiment (**right column**). $\psi(t)$, $\psi^*(t)$, $\psi_M(t)$ (**upper plot**) and IMR discrepancy signal $\sigma(t)$ (**lower plot**) time histories for $\psi^* = \pi \sin(0.1t)$. Sine-modification.

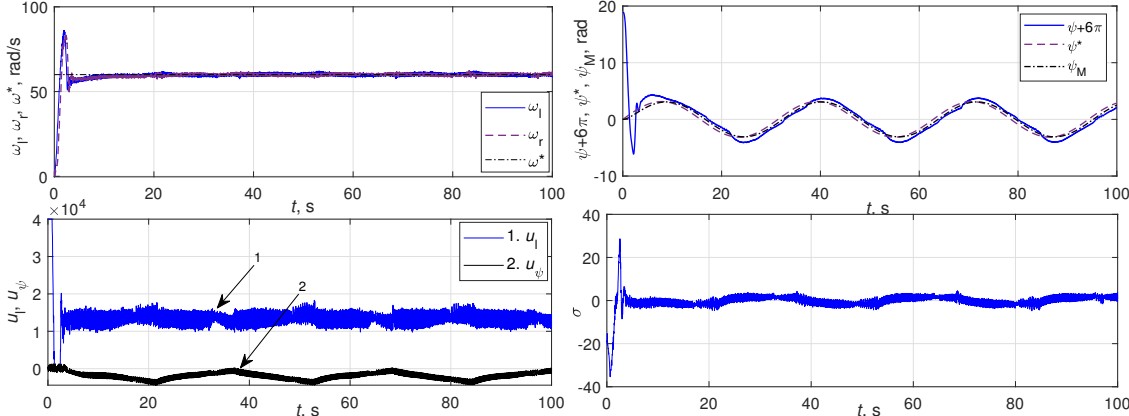

**Figure 11.** Experiment. Left column: $\omega_l(t)$, $\omega_r(t)$, $\omega^*(t)$ (**upper plot**) and control actions $u_l(t)$, $u_\psi(t)$ (**lower plot**) time histories; right column: $\psi(t)$, $\psi^*(t)$, $\psi_M(t)$ (**upper plot**) and IMR discrepancy signal $\sigma(t)$ (**lower plot**) for $\psi^* = \pi \sin(0.2t)$. Sine-modification.

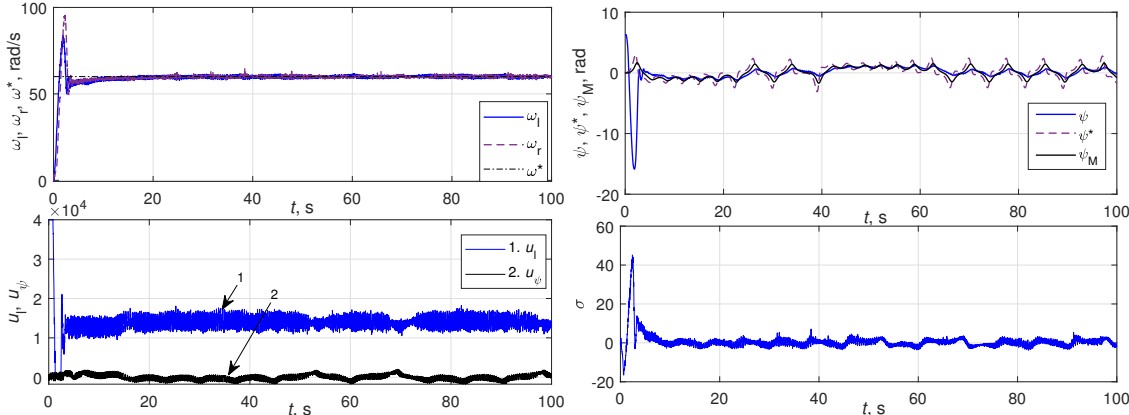

**Figure 12.** Experiment. Left column: $\omega_l(t)$, $\omega_r(t)$, $\omega^*(t)$ (**upper plot**) and control actions $u_l(t)$, $u_\psi(t)$ (**lower plot**) time histories; right column: $\psi(t)$, $\psi^*(t)$, $\psi_M(t)$ (**upper plot**) and IMR discrepancy signal $\sigma(t)$ (**lower plot**) time histories for chaotic $\psi^*(t)$ (51), $m_t = 0.1$, $c = 0.125$. Sine-modification.

### 5.4. Limitations of the Proposed Solution

In Section 3.2, the model of mechatronic stand dynamics is discussed. Its complexity, the difficulties in its complete description are outlined, and simplifying assumptions that were used in the synthesis and numerical study (computer simulation) of the proposed control algorithm are mentioned. Actually, in a real system, the efficiency of the proposed algorithm is limited, and these limitations cannot be revealed by the simplified model used. The effect of these restrictions is manifested depending on the operating speeds of the drives.

1. At rotor speeds of about 23 rad/s, there is a lower resonant frequency of movement of the main stand platform. In this case, "destructive" oscillations of the platform appear, which come into collision with the stand supports. Naturally, imparting the required constant speed and phase of the rotors' rotation is impossible due to the significant mechanical connection between the movements of the platform and the rotors revolving.

2. The rotation speed must be high enough to ensure the averaging property specified in Section 3.2 , cf. [1,39,49]. If this is not the case, then the gravitational ("pendular") torque acting on the unbalanced rotors is not averaged during rotation but leads to significant fluctuations in the speed of the rotors revolving. This effect cannot be smoothened by the rotor speed control algorithm used, and respectively, it is not possible to provide the required phase shift between the rotors' angles. As shown in [49], and confirmed by the present study, the lower frequency bound where the averaging property manifests itself is about 30 rad/s. The experimentally obtained time histories of $\omega_l(t)$, $\omega_r(t)$, $\psi(t)$, $\psi^*(t)$ for $\omega^* = 10$ rad/s, $\psi^* = 0$ are shown in Figure 13.

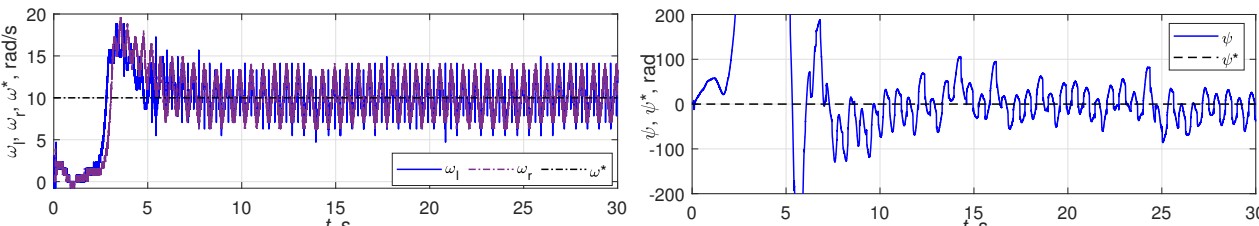

**Figure 13.** Experiment. **Left plot**: $\omega_l(t)$, $\omega_r(t)$, $\omega^*(t)$; **right plot**: $\psi(t)$, $\psi^*(t)$ time histories for $\omega^* = 10$ rad/s, $\psi^* = 0$.

3. If the revolving speed of the rotors is sufficiently high, then the phenomenon of self-synchronization of their rotation acquires a significant influence. This effect is essential for oscillatory systems and was first noticed by C. Huygens. It was described in detail in the works by I. Blekhman and his colleagues, cf. [1,7,10,41–45]. Revolving imbalanced rotors are subject to mutual influence through "pushes" transmitted between the rotors by the stand structure parts. Despite the fact that the proposed algorithm is able to provide the given rotation speeds of each rotor, it is not possible to obtain the required phase shift between their angles. An experimental study of this phenomenon is presented in [49] and confirmed by the present research. For the high-frequencies, starting from 75 rad/s, the achievable phase shift range is narrowed, but if the frequency is 125 rad/s, then the desired phase shift up to $\pm \pi/2$ rad can be ensured. This property is demonstrated by the experimental results shown in Figure 14, where the time histories of $\omega_l(t)$, $\omega_r(t)$, $\psi(t)$ for $\omega^* = 80$ rad/s, $\psi^* = \pi$ are plotted. It is seen that $\psi(t)$ does not tend to the desired value $\pi$ but demonstrates slow oscillations around it with those close to $\pi$ radians magnitude.

4. At the rotation frequency close to 125 rad/s, the Sommerfeld effect [1] takes place. Its appearance prevents a further increase in the speed of the rotors revolving, and special measures are required to overcome this effect. The proposed algorithms do not work under these circumstances.

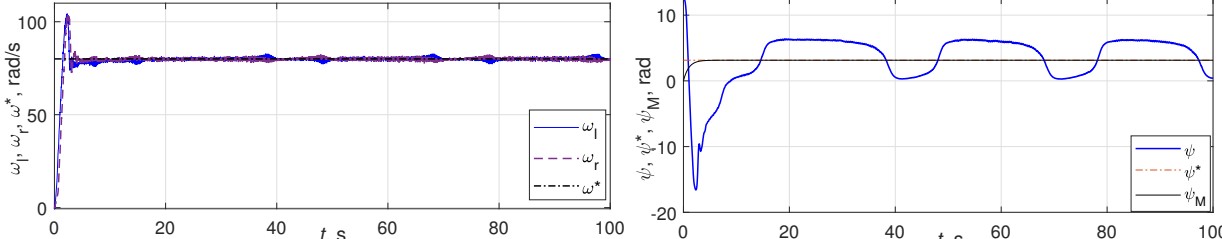

**Figure 14.** Experiment. **Left plot**: $\omega_l(t)$, $\omega_r(t)$, $\omega^*(t)$; **right plot**: $\psi(t)$, $\psi^*(t)$ time histories for $\omega^* = 80 \, \text{rad/s}$, $\psi^* = \pi$.

Thus, for this stand, the operating range of using the proposed algorithm is 30–75 rad/s. Note that the low-frequency region is not typical for vibration technologies; the main interest is in the upper frequencies.

From the point of view of the authors, these limitations are inevitably inherent in the physical system under consideration; therefore, the use of more complex models, as, for example, in [63], for designing a controller can hardly lead to an expansion of control capabilities, since it is intuitively clear that feedback over the control loop is rather "soft" due to transients in electromagnetic circuits of controller equipment, whereas a direct rigid mechanical connection between the rotors is provided by the expense of structural interrelation.

*5.5. Robustness Analysis*

The proposed control algorithm (39)–(48) was positioned as a robust one. This property is provided in it by the presence of relay components, in contrast to the control law of [11], where robustness is carried out by adaptive adjustment of the controller parameters. This property is numerically examined by simulations.

The following objective functions are chosen: (1) the control error settling time $t_e^*$, that is, the moment from which the error $e_\psi(t)$ falls into the 3% zone from its initial value $e_\psi(0)$ and remains in it; (2) time $t_\sigma^*$ taken for discrepancy signal $\sigma(t)$ in (45) finally entering the 3%-zone of $\sigma(0)$. This criterion is reasonable since, as noted above, for systems with IRM, the proximity of $\sigma(t)$ to zero indicates that the closed-loop system dynamics are close to the reference ones.

Below, the examination results of the deviating plant model parameters on ±50% from the initial ones, which can be called "nominal", are derived by the identification procedure of Section 3.3 as $b_0 = 0.0042 \, \text{s}^{-1}$, $a_0 = 0.119 \, \text{s}^2$, $a_l = 0.811 \, \text{s}$. Control law (39)–(48) gains were taken as $\gamma = 1000$, $\gamma_I = 500$.

The dependence of quality indices $t_e^*$, $t_\sigma^*$ on the control law gains $\gamma \in [0, 2000]$, $\gamma_I \in [0, 1000]$ is studied for the fixed (nominal) plant model parameters. This study is aimed not only at establishing the robustness of the control algorithm but also at indicating how sensitive the system performance is to the choice of controller parameters during the design.

Throughout the studies, the parameters of the PI speed controllers (39)–(41) were taken from [10] as $k_I = 240 \, \text{s}$, $k_P = 1680$, for both right, and left, velocity controllers. The calculations were carried out for a fixed reference frequency $\omega^* = 60 \, \text{rad/s}$ and a reference phase shift as $\psi^* = \pi \, \text{rad}$, and the sampling interval was set to $T_s = 0.02 \, \text{s}$.

5.5.1. Robustness with Respect to Plant Model Parameters

The numerical study results as dependencies of $t_e^*$, $t_\sigma^*$ on plant model parameters $a_1$, $b_0$ are demonstrated by the 3D plot and the contour chart in Figures 15 and 16, respectively. It is seen that the plant parameter variations in the wide range do not lead to a drastic change in the close-loop system behavior and, therefore, confirm its robustness. Furthermore, the simulation results show a characteristic property of systems with the IRM: the decay time $t_\sigma^*$ of the discrepancy signal $\sigma(t)$ in the control algorithm turns out to be significantly less

than the time of the transient process $t_e^*$ in the main control loop, that is, the dynamics of the controlled variable reach the desired one even during the transients.

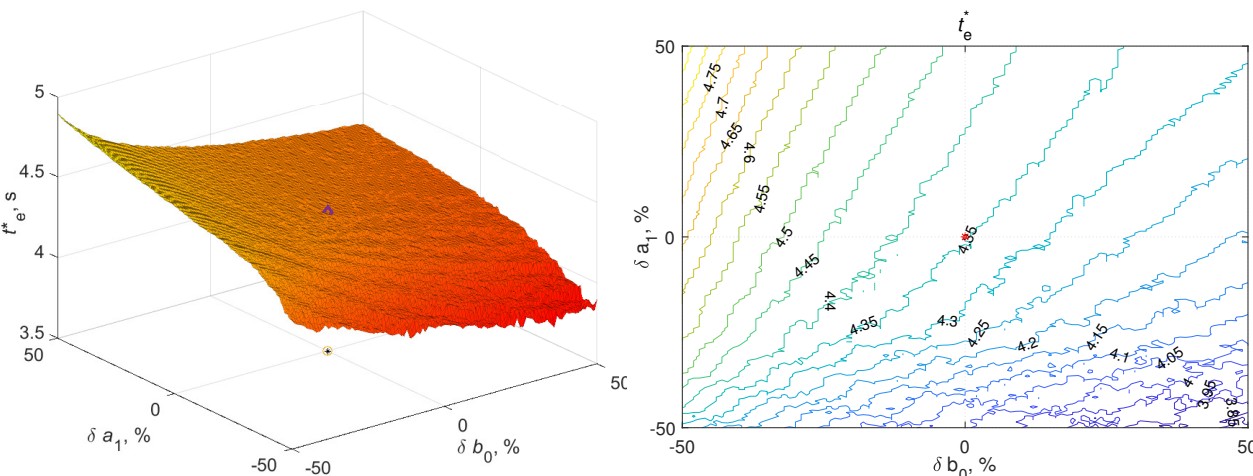

**Figure 15.** Simulations results. Dependence of $t_e^*$ on plant model parameters $a_1$, $b_0$, varying on $\pm 50$ %; $\gamma = 1000$, $\gamma_I = 500$, $\omega^* = 60$ rad/s, $\psi^* = \pi$. **Left**: 3D plot, **right**: contour plot; asterisk $\star$ is for "nominal" parameters.

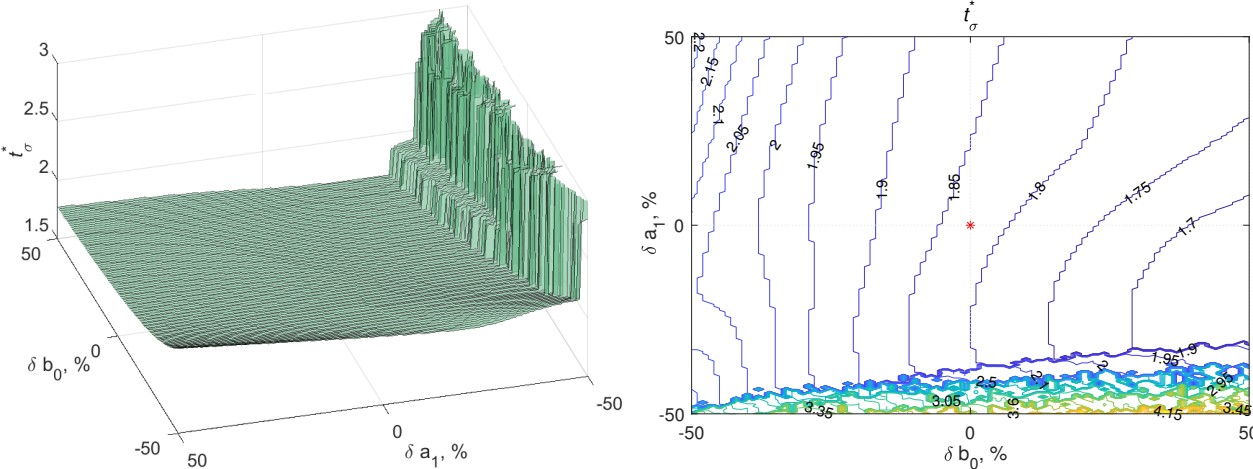

**Figure 16.** Simulations results. Dependence of $t_\sigma^*$ on plant model parameters $a_1$, $b_0$, varying on $\pm 50$ %; $\gamma = 1000$, $\gamma_I = 500$, $\omega^* = 60$ rad/s, $\psi^* = \pi$. **Left**: 3D plot, **right**: contour plot; asterisk $\star$ is for "nominal" parameters.

5.5.2. Robustness with Respect to Controller Parameters

The results of the numerical analysis of the system robustness with respect to controller coefficients are shown in Figures 17 and 18 in the form of a 3D plot and a contour chart. It can be seen that the surfaces of indices $t_e^*$, $t_\sigma^*$ are rather "flat"; thus, the procedure for controller design does not require a deep analysis of the model and calculations, as it is enough to choose them, as it did do, based on reasonable grounds based on a preliminary acquaintance with the properties of the object under consideration as a result of several experiments. The graphs also show a sharp increase in quality indicators at some borders. It is associated with the sensitivity of the selected criteria to the phenomenon of chattering, which manifests itself in the region of small values of the coefficient $\gamma$ of direct (relay-proportional) connection compared to the relay-integral gain $\gamma_I$. The chattering amplitude itself is small (and is less for system output compared with auxiliary signal $\sigma$), but in this region, it may be outside the 3% zone for which the process durations are calculated.

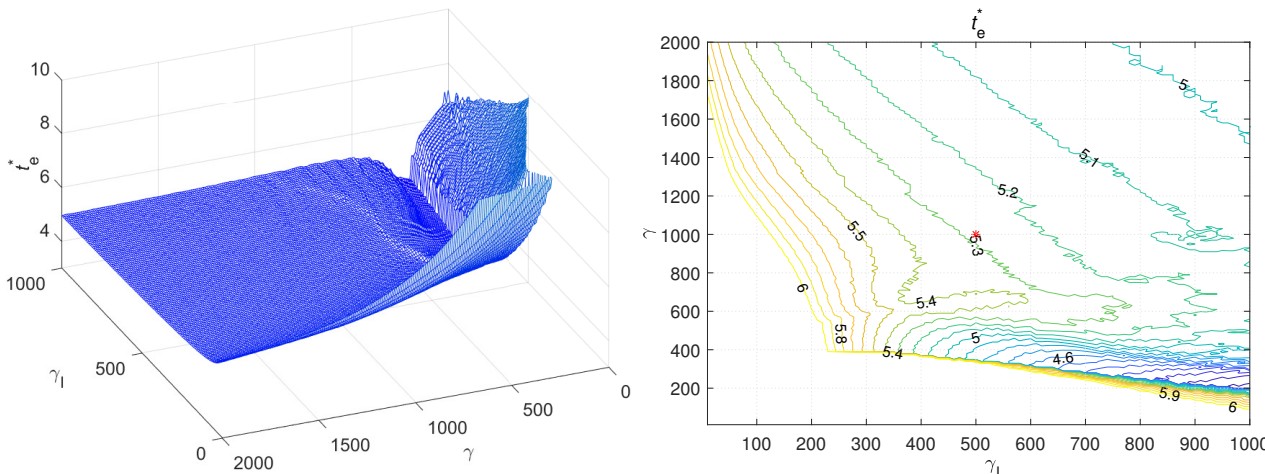

**Figure 17.** Simulations results. Dependence of $t_e^*$ on controller gains $\gamma$, $\gamma_i$ for "nominal" plant parameters; $\omega^* = 60$ rad/s, $\psi^* = \pi$. **Left**: 3D plot, **right**: contour plot; asterisk $\star$ is for $\gamma = 1000$, $\gamma_I = 500$.

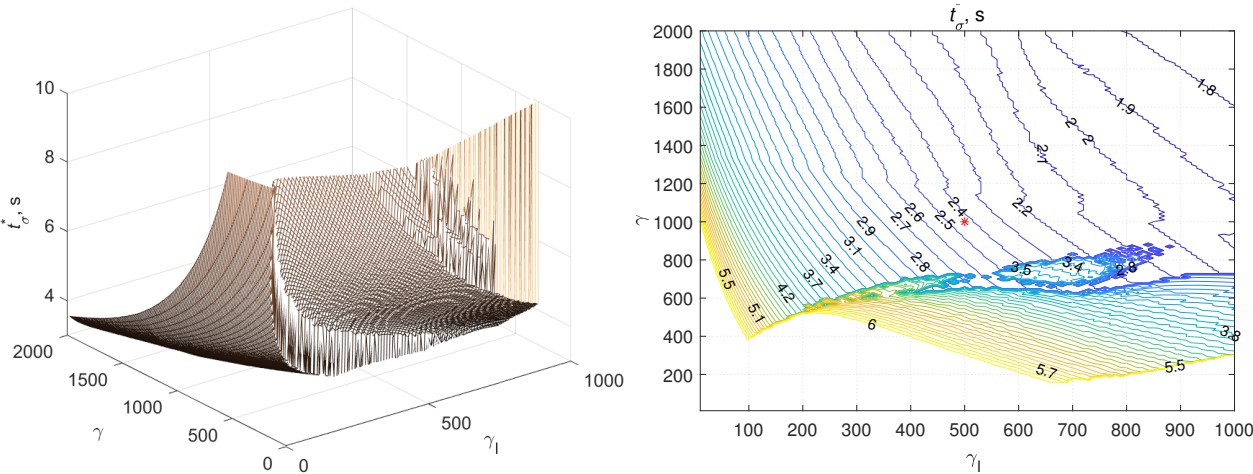

**Figure 18.** Simulations results. Dependence of $t_\sigma^*$ on controller gains $\gamma$, $\gamma_i$ for "nominal" plant parameters; $\omega^* = 60$ rad/s, $\psi^* = \pi$. **Left**: 3D plot, **right**: contour plot; asterisk $\star$ is for $\gamma = 1000$, $\gamma_I = 500$.

## 6. Conclusions

In the paper, the problem of robust control of the phase shift during rotation at a given speed of the unbalanced rotors for a two-rotor vibratory machine is presented by designing the controller, including two proportional-integral (PI) rotor speed controllers with a cross-coupling, and a relay-type signal controller with an integral component. For the control law design, the speed-gradient method was employed. For various types of reference phase shift signals (constant, harmonic, chaotic), the results of extensive experimental studies, performed on the mechatronic vibration setup and the simulations, accomplished based on the results of identifying the parameters of the stand drive model are presented in the paper. The obtained results confirm the efficiency of the proposed algorithm and allow one to reveal the system performance properties. The main results of the paper are consolidated in Table 1.

In future works, we will use the new stability analysis results to prove that the other parameters do not just "remain bounded", which may also imply that they may keep moving around at their free will but that they actually end at finite values. In the future, it is also planned to consider the possibility of replacing PI speed controllers with relay ones similar to those used in this work in the phase control loop. It seems possible to circumvent

the need to measure the angular acceleration and the difficulties associated with this by using a parallel feedforward compensator (a "shunt").

**Table 1.** Main results.

| # | Content | Comment |
|---|---------|---------|
| 1 | Simplification of stability analysis for eliminating certain continuity conditions | Section 2.1, Theorem 1 |
| 2 | Relay phase-shift control law with IRM | Equations (39)–(41) |
| 3 | Sine-modification of phase-shift controller | Equation (49) |
| 4 | Algorithm for parameteric identification of drive systems | Section 3.3 |
| 5 | Analysis of the limiting possibilities of feedback synchronization control for two-rotor vibrating machines | Section 5.4 |
| 6 | Robustness analysis of relay phase-shift control | Section 5.5 |

**Author Contributions:** Conceptualization, B.A.; data curation, I.Z.; formal analysis, I.B. and I.Z.; funding acquisition, B.A.; investigation, I.Z.; methodology, B.A.; project administration, B.A.; software, I.Z.; supervision, B.A.; writing—original draft, B.A. and I.B.; writing—review and editing, B.A. and I.B. All authors have read and agreed to the published version of the manuscript.

**Funding:** This research received no external funding.

**Data Availability Statement:** Not applicable.

**Acknowledgments:** The paper is dedicated to the blessed memory of Dmitry Indeitsev (December 1948–December 2022), who headed the IPME RAS for many years—a wonderful person, scientist, and teacher. The authors are grateful to Alexander L. Fradkov for the helpful ideas behind the paper and Vladimir I. Boikov for his invaluable work in creating the electronic and computer facilities for the MMLS SV-2M.

**Conflicts of Interest:** The authors declare no conflict of interest.

## Abbreviations

The following abbreviations and notations are used in this manuscript:

| | |
|---|---|
| DoF | Degrees of Freedom |
| FC | Frequency Converter |
| FIR | Finite Impulse Response |
| IM | Induction Motor |
| IRM | Implicit Reference Model |
| LSE | Least-Square Estimation |
| MMLS | Multi-Resonance Mechatronic Laboratory Setup |
| MRAC | Model Reference Adaptive Control |
| PD | Proportional-Differential |
| SAC | Simple Adaptive Control |
| SG | Speed Gradient |
| SGA | Speed-Gradient Algorithm |
| $\Gamma^{\dagger}$ | pseudo-inverse to matrix $\Gamma$ |

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
