# Peer review of "Passification-Based Robust Phase-Shift Control for Two-Rotor Vibration Machine"

_electronics, doi:10.3390/electronics12041006_

Round 1

Reviewer 1 Report

In this paper, the passification-based robust phase shift control for two-rotor vibration machine is studied. And the solution to the problem of robust control of the phase shift during rotation at a given speed of the unbalanced rotors is presented. The content of the paper is rich and the theory description is strong. Many prospects are put forward in the end of the Conclusion part.    It sounds good that the simulations results are agreeable with the presented algorithm.   The article is well organized and well written. However, there are some quenstions as follows,(1) Authors can give some more descriptions and related works on ‘Speed-Gradient Method’ in Part 2.2. (2)  In Fig. 2, the setup control cannot be easily seen,   redraw the  functional electrical diagram and mark the control flow. (3) Please unify the reference format so that the DOI number can be appended to each article.  (4)  There are some grammatical mistakes in the paper, the authors can check carefully. 

Author Response

Reviewer #1
In this paper, the passification-based robust phase shift control for two-rotor vibration machine is studied. And the solution to
the problem of robust control of the phase shift during rotation at a given speed of the unbalanced rotors is presented. The content of
the paper is rich and the theory description is strong. Many prospects are put forward in the end of the Conclusion part. It sounds
good that the simulations results are agreeable with the presented algorithm. The article is well organized and well written.
Authors’ response
Thank you for the positive, in general, assessment of the paper, the useful remarks, and the suggestions. In the
revised manuscript all your comments are taken into account. The manuscript is improved accordingly; the changes in it
are highlighted in blue.
However, there are some quenstions as follows,
(1) Authors can give some more descriptions and related works on ‘Speed-Gradient Method’ in Part 2.2.
Authors’ response
Thanks for the recommendation. This method is widely presented in the literature, including reviews, references
to which are available in the article. In order not to increase the amount of material that is not directly related to this
article (which caused a comment by another reviewer) and to increase the number of self-citations (which also caused a
comment), in this article we limited ourselves to adding a brief explanation at the beginning of the paragraph (highlighted
in blue in the paper):
At the end of 1970-th it turned out to be possible that a unification of various control algorithms can be achieved if
the gradient of the rate of objective function change along the trajectories of the controlled plant is employed. Apparently,
the most general scheme for constructing algorithms was proposed in [14]. The resulting algorithms were called
speed-gradient (SG) algorithms.
(2) In Fig. 2, the setup control cannot be easily seen, redraw the functional electrical diagram and mark the control flow.
Authors’ response
Thank you for the suggestion. The functional electrical diagram for setup control, Fig. 2. was redrawn marking the
control flow.
(3) Please unify the reference format so that the DOI number can be appended to each article.
Authors’ response
Thank you for the suggestion. DOI numbers were appended wherever it was possible.
(4) There are some grammatical mistakes in the paper, the authors can check carefully.
Authors’ response
Thank you for pointing out this issue. The errata and mistakes in English spelling and grammar were double-checked
throughout the paper and fixed.

Reviewer 2 Report

The work presents a controller design for nonlinear systems, applied to the control of a two-rotor vibration machine. Some issues are present in the work:

1. The contribution of the work over existing schemes is not immediately apparent from the Introduction. Specifically, while the authors cited many different works (78 in total), it is unclear what improvements are made in this paper over the existing results.

2. The meaning of lines 66--67 (`Changing the parameters... of the vibration drives.') is not clear.

3. The explanation in subsection 2.1 is excessive. Please only cite the relevant theorems/assumption, and summarise the findings. The paper should focus on the mathematical constructs relevant to the application, not go excessively detailed into the history of how the final theorem that is applied came to be. Also provide the appropriate citations for material taken from existing results.

4. Proofread the equation numbering in lines 374 and 379. Also spellcheck the paper for spelling mistakes (e.g., `imlicitly' in line 394).

5. Verify that Assumption 1 is satisfied by the example system.

6. The authors also cite their own work excessively; 21/74 of the references are by the authors!

Author Response

Reviewer #2
The work presents a controller design for nonlinear systems, applied to the control of a two-rotor vibration machine. Some issues
are present in the work:
Authors’ response
Thank you for the useful remarks, and the suggestions. In the revised manuscript all your comments are taken into
account. The manuscript is improved accordingly; the changes in it are highlighted in blue.
1. The contribution of the work over existing schemes is not immediately apparent from the Introduction. Specifically, while the
authors cited many different works (78 in total), it is unclear what improvements are made in this paper over the existing results.
Authors’ response
Thanks for the comment. Of course, not all works from the citation list relate directly to the considered problem of
controlling a vibration stand, but indeed, there are similar works. In many existing publications, a control algorithm is
presented, but the results are not confirmed by field experiments, but only by simulations. There are reasons to believe
that in many of the publications, the algorithm will not be confirmed due to the structural discrepancy between the
system under consideration and the model (for more details, please see section Sec. 3.2). The closest to this study is work
[11]. The present publication is continuation in the direction of simplification, and making the control algorithm more
robust and faster, which is achieved by refusing to use the parametric adaptation, and switching to the signal algorithm.
The authors do not want to make unnecessary criticisms of other publications, so only the following remarks have been
added to the introductory part (highlighted in blue):
In many existing publications, the results are not confirmed by experiments, but only by simulations so the structural
discrepancy between the system under consideration and its model can easily lead to unattainability in the practice of
the posed control goal. The closest to this study is work [11], and the present publication extends it to the direction of
simplification, making the control algorithm more robust and faster, which is achieved by refusing to use the parametric
adaptation, and switching to the signal algorithm.
In this paper, we also use new results that simplify stability analysis and eliminate some demanding continuity
conditions that are usually considered to be necessary for the stability of the system under analysis. Although most
publications base their stability analysis on Barbalat’s lemma which seems to make continuity the necessary condition,
recent publications present a new theorem of stability which is a direct extension of the original Lyapunov theorem
of stability for the case when the Lyapunov derivative is only negative semidefinite, and they managed to show that
continuity is not needed for stability.
2. The meaning of lines 66–67 (‘Changing the parameters... of the vibration drives.’) is not clear.
Authors’ response
The following clarified sentence is introduced instead of the mentioned one in the paper:
Changing the vibro-transportation parameters is traditionally carried out using the settings of the vibration drives,
for example, by the hand-made adjustment of the eccentric masses positions.
3. The explanation in subsection 2.1 is excessive. Please only cite the relevant theorems/assumption, and summarise the findings.
The paper should focus on the mathematical constructs relevant to the application, not go excessively detailed into the history of how
the final theorem that is applied came to be. Also provide the appropriate citations for material taken from existing results.
Authors’ response
As we mentioned, like most publications, even the main authors of this paper would have started their stability
analysis with the commonly used Barbalat’s Lemma and with the common condition that everything must be continuous.
Therefore, because we are referring to a new direction in stability analysis, we thought to give the due respect to those
works of LaSalle and Artztein who started the new trend. We would be glad to know that the reviewers now the referred
publications, in particular LaSalle’s publication, which then allowed simplifying and extending the stability analysis of
nonlinear systems. Our experience taught us that when people refer to LaSalle, they mean the 1950-1960 works and not
the 1976-1980 publications that we found it right to mention because they have remained surprisingly unknown and
are not even mentioned in the best nonlinear system books. Those new results change what one can read in almost all
publications and therefore have been published only after “natural” rejections and only because reviewers were able to
also ask specific questions, then to read explanations, and then to finally end changing their initially negative opinion.
Cutting the (in our humble opinion, not too large) explanation would lead the reader directly to the new Theorem and
leave many questions unanswered, and regular readers may have problems dealing directly with the ultimate result.
We hope that the reviewer will allow this explanation of the developments that ultimately led to the new Theorem of
Stability.
4. Proofread the equation numbering in lines 374 and 379. Also spellcheck the paper for spelling mistakes (e.g., ‘imlicitly’ in
line 394).
Authors’ response.
Thank you for pointing out this inaccuracy. It has been corrected as follows:
The phase shift control law (37)–(39), as a part of the common control algorithm (34)–(40),...
In the present study, phase shift control algorithm (34)–(40)...
The errata and mistakes in English spelling and grammar were double-checked throughout the paper and fixed.
Particularly, the mentioned paragraph in the new version of the manuscript is changed to:
Version January 30, 2023 submitted to Electronics 3 of 4
...is not a part of the system, but implicitly represented by parameter tM of the algorithm. Based on this property,
this method is called the “IRM method”. Although signal yM is not used in the control law (34)–(50), it is shown, for
clarity, together with y(t) time histories in the experimental part, see Sec. 5.
5. Verify that Assumption 1 is satisfied by the example system.
Authors’ response Thank you for the suggestion.
As we already wrote, Assumption 1 is satisfied in all cases when bounded trajectories cannot pass an infinite distance
in finite time. How could this Assumption be violated? Even if the differential equation contains impulses, they would
only lead to bounded jumps of the trajectories. Could a sequence of impulse functions make these bounded jumps sum
to an infinite distance in finite time? It is easy to see that this could occur only if there are an infinitely dense sequence of
impulse functions in a finite interval. As this is improbable in any realistic plant, Assumption 1 is usually satisfied.
The clarifying remark (Remark 2, page 13) is added to the paper.
6. The authors also cite their own work excessively; 21/74 of the references are by the authors!
Authors’ response
Thank you for the suggestion. In the revised manuscript the number of self-citations is redused to 15, which gives
15/64 = 23%.

Reviewer 3 Report

After careful reading the paper, it is a good direction in this field, I have some suggestions.

1. Abstract: Keywords are not matching, such as: robust control, vibration technologies, and adaptive controller, the author seems do not catch major point. Please recheck.

2. The major theme of the paper is to discuss robustness, but most of the paper uses the derivation of mathematical models, and using case study to verify the creative point that the author presented, and the results all are appear in the Figure.

If the author could, please uses Table and cardinal values to show the results that the paper want to give reader..

3. In conclusion, please add the limitation of the research.

4. In page 18, line497, Ref[1]] and page 19, REF[16], REF[17], REF[18], REF[21], EF[22], REF[30], REF[31], REF[32], REF[33], REF[34], REF[35] and REF[36] are too far away, is the Bible of this field?

Author Response

Reviewer #3
After careful reading the paper, it is a good direction in this field, I have some suggestions.
Authors’ response
Thank you for the positive, in general, assessment of the paper, the useful remarks, and the suggestions. In the
revised manuscript all your comments are taken into account. The manuscript is improved accordingly; the changes in it
are highlighted in blue.
1. Abstract: Keywords are not matching, such as: robust control, vibration technologies, and adaptive controller, the author
seems do not catch major point. Please recheck.
Authors’ response
Based on the Reviewer’s suggestion, the keywords were changed to:
vibration machine; unbalanced rotors; mechatronics; phase shift; speed-gradient; relay control; robustness; nonlinear
control
2. The major theme of the paper is to discuss robustness, but most of the paper uses the derivation of mathematical models, and
using case study to verify the creative point that the author presented, and the results all appear in the Figure.
Authors’ response
Thank you for the useful and inspiring suggestion. Based on it, we added to the paper Sec. 5.5 “Robustness
Analysis”, where the proposed control algorithm robustness is numerically studied and demonstrated.
If the author could, please uses Table and cardinal values to show the results that the paper want to give reader.
Authors’ response
Based on the Reviever’s suggestion, the main results of the paper are consolidated in Conclusion, Tab. 1.
3. In conclusion, please add the limitation of the research.
Authors’ response
Thank you for the inspiring suggestion. The mentioned issue is highly important for the subject under the consideration,
hence it encouraged us to add, starting at page 17, a special subsection with thorough explanations, including new
experimental results, see Sec. 5.4 of the revised manuscript.
4. In page 18, line497, Ref[1]] and page 19, REF[16], REF[17], REF[18], REF[21], EF[22], REF[30], REF[31], REF[32],
REF[33], REF[34], REF[35] and REF[36] are too far away, is the Bible of this field?
Authors’ response
Based on the Reviever’s suggestion, some less relevant citations were dropped in the revised manuscript, namely
(numbering is given as in the original version): Ref[1], REF[32], REF[33], REF[34], REF[36], from the Reviever’s list, and
Ref [10], Ref [11], Ref [44], Ref[45], Ref[46], Ref[48], Ref[49] for reducing the number of self-citations. The other references
seem to be essential for the paper’s context. The total number of citations is reduced from 75 to 65.

Round 2

Reviewer 2 Report

The authors have addressed many of my comments well. It is still very unclear if such a long introduction is required when the points can be summarised into briefer paragraphs. Nevertheless, the points the authors tried to make get across somewhat.

More importantly, the explicit theoretical contribution of the work is in extending the results in [11]. It would be more helpful if the introduction of the new approach also discussed other pervious results, so it is easier to compare the new approach with existing results and explain its novelty/strengths over previous limitations.

Author Response

Reviewer #2
The authors have addressed many of my comments well. It is still very unclear if such a long introduction is required when the
points can be summarised into briefer paragraphs. Nevertheless, the points the authors tried to make get across somewhat.

Authors’ response
Thank you for the suggestion. In the revised manuscript, the introduction section has been substantially shortened,
and many less important references have been removed.
More importantly, the explicit theoretical contribution of the work is in extending the results in [11]. It would be more helpful if
the introduction of the new approach also discussed other pervious results, so it is easier to compare the new approach with existing
results and explain its novelty/strengths over previous limitations.

Authors’ response
Thank you for the suggestion. The following clarifying paragraph is added to Introduction (highlighted in blue):

The approach of [9,10] has been extended by implementing simple adaptive control (SAC) with an implicit reference
model (IRM) in [11], where the possibility of efficiently applying the SAC method to a real technical system is shown.
The algorithm obtained in [11] refers to the velocity gradient (SG) algorithms, cf. [14–18], and Sec. 2.2 below, with the
implementation of the controller parameters tuning. However, in real problems, in the presence of unmodeled dynamics,
disturbances (for example, caused by the platform vibrations), noises, measurement errors, and data sampling, the
parameters of the adjustable controller can leave the allowable zone. Reducing the gain of the adaptation algorithm
in combination with parametric feedback in it, as well as anti-windup correction, reduce the achievable speed of the system, and its accuracy in the tracking mode. Therefore, in this work, another type of SG algorithms is developed and studied, namely, a robust signal algorithm, where adjustment of the controller parameters does not employ. This approach allows, while maintaining robustness, to increase the speed and accuracy of the control process while avoiding a possible  divergence of the controller parameters.

The Authors,
Boris Andreivsky
Iuliia Zaitceva
Itzhak Barkana

Round 3

Reviewer 2 Report

My queries have been addressed sufficiently.